# ADDING RECURRENCE TO PRETRAINED TRANSFORMERS

## ABSTRACT

Fine-tuning a pretrained transformer for a downstream task has become a standard method in NLP in the last few years. While the results from these models are impressive, applying them can be extremely computationally expensive, as is pretraining new models with the latest architectures. We present a novel method for applying pretrained transformer language models which lowers their memory requirement both at training and inference time. An additional benefit is that our method removes the fixed context size constraint that most transformer models have, allowing for more flexible use. When applied to the GPT-2 language model, we find that our method attains better perplexity than an unmodified GPT-2 model on the PG-19 and WikiText-103 corpora, for a given amount of computation or memory.

## 1 INTRODUCTION

Recent progress in NLP has been dominated by large pretrained transformer neural networks (Vaswani et al., 2017), such as BERT (Devlin et al., 2019), and GPT-2 (Radford et al., 2019). However, these models have a memory footprint that is quadratic in input sequence length. Although architectural innovations such as those of Kitaev et al. (2019) and Rae et al. (2019) mitigate this and the issue of a predetermined maximum context size, large pretrained models applying these techniques are not available at this time. Even if large pretrained models of this kind are released in the future, they will likely not cover the wide range of domains that BERT-family models have been published for. For example, there have been BERT-based models trained for other languages such as French (Le et al., 2020; Martin et al., 2020), Italian (Polignano et al., 2019), and many other languages (see Nozza et al. (2020) for an overview) as well as specific domains such as scientific papers (Beltagy et al., 2019), biomedical papers (Lee et al., 2020), and health records (Rasmy et al., 2020). Individuals working with these models may not have the resources to train new models from scratch using the latest tricks, as the computation requirements for pretraining are extremely high. As such, identifying ways that already existing models can be improved could be widely impactful.

Another drawback of this family of models is that they have an a priori fixed maximum context size (typically 512 or 1024 tokens for the currently available pretrained models). A typical application of pretrained language models is producing contextual embeddings for a document. If the document is simply chunked into disjoint segments of 512 tokens, tokens at the boundary of a window will have less contextual information than tokens in the center of a window. This can be mitigated by striding the evaluation of the model, and only keeping the embedding for a token which has the largest context—but this adds quite a bit of wasted computation.

In this paper, we propose a method for augmenting and fine-tuning pretrained transformer language models to use context without directly attending to it. Our method simultaneously allows for increasing the context size a transformer processes, while allowing a controllable trade-off between computation and perplexity. We accomplish this by adding a small recurrence module that computes a fixed size representation from the transformer hidden states in a window of text. Then, the representation for that window is used during processing of the next window. Shrinking the window size is then a way to reduce the memory footprint of the model, with less loss of performance than would occur with a standard transformer. Our experiments add recurrence GPT-2 language models, and fine-tune them on the PG-19 (Rae et al., 2019) and WikiText-103 corpora (Merity et al., 2016), and require only the same amount of memory used for standard fine-tuning of a pretrained

language model. We demonstrate improvements in perplexity compared to a baseline model using the same amount of computation. Qualitative analysis shows that our recurrent module propagates certain information from previous windows of text, which can facilitate handling of long-distance dependencies with fixed-size input windows.

## 2 RELATED WORK

Many methods have been proposed to lower the memory footprint or computation time of transformer language models, or allow them to be used on larger contexts. The Transformer-XL (Dai et al., 2019) allows a position within an attention window to attend to tokens from the previous windows by introducing relative position embeddings. While that mechanism, like ours, allows information to flow between windows, existing BERT and GPT-2 models do not use relative position embeddings, so training from scratch would be necessary to take advantage of this architecture. Additionally, each layer in the Transformer-XL attends to the previous layer in the previous window, so the maximum attention horizon is finite. Our recurrent method could theoretically pass information across an arbitrary distance, although one would not expect it to exceed the Transformer-XL's horizon without a much larger scale of data than we experiment with.

We list here some other modifications of the transformer architecture, somewhat imprecisely grouping them for brevity. For a more detailed discussion, see Tay et al. (2020b). Child et al. (2019), Qiu et al. (2019), Kitaev et al. (2019), Sukhbaatar et al. (2019), and Roy et al. (2020) introduce sparsity to self-attention in various forms, reducing its memory cost. Rae et al. (2019) and Beltagy et al. (2020)—dynamically and statically respectively—add extra tokens to attend to which allow for global passing of information. Tay et al. (2020a) and Wu et al. (2019) replace dynamically computed self-attention with cheaper alternatives. While the above methods all allow for a reduction in computation, they also all require training from scratch. Our goal is to allow more efficient and powerful use of the wide array of existing pre-trained models that cover many domains.

Cao et al. (2020) propose the DeFormer, which also modifies the execution of a pretrained transformer. However, unlike our method, they decompose a single window into multiple windows by removing the attention interactions between these windows. This is largely orthogonal to our method, as one could both decompose windows of text, and additionally use our method to allow information to be passed between neighboring windows. Similarly, distilled versions of pre-trained models such as DistilBERT (Sanh et al., 2019) provide more computational efficiency, but could be combined with our method to apply them to longer contexts, or reduce the quadratic cost of self-attention.

Hao et al. (2019) apply pre-trained transformers recurrently for machine translation, but do so by using an attention network to embed the document, applying a recurrent encoder to those embeddings, and using the recurrent encoder alongside a typical transformer encoder. This differs from our method as we are fine-tuning language models, which are transformer decoders, and directly modifying the transformer's computation with a recurrent connection, rather than running an RNN on top of embeddings produced by a transformer.

## 3 METHOD

The main idea of our method is to take a transformer that was pretrained in a fixed context size setting and add recurrence at the level of $T$-token windows of text. For example, instead of executing the model on one 1000 token window of text, we could instead execute our model with 10 windows of 100 tokens. The first window is processed by the transformer model as normal, but for subsequent windows we add a supplementary embedding, which is generated using the hidden states from the preceding window (see Figure 1). The recurrence module is extremely small compared to the size of transformer language model, so the additional computation required is negligible.

### 3.1 ADDING RECURRENCE TO PRETRAINED TRANSFORMERS

Starting by defining terms, we will consider a pretrained transformer with $L$ layers, a hidden state size of $k$, and a maximum context size of $T$ tokens. Let $\boldsymbol{h}_i^{(\ell)} \in \mathbb{R}^k$ be the output of the $\ell$-th layer of the pretrained model, at position $i$. To produce a fixed-size representation of tokens $t_1, t_2, \ldots, t_T$,

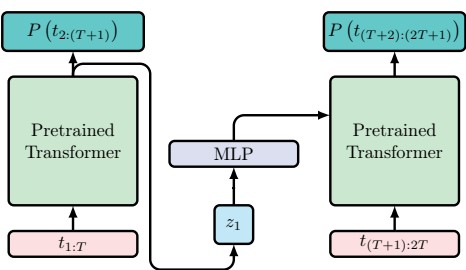
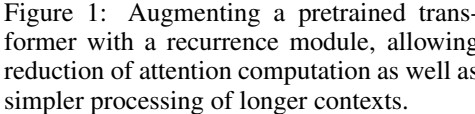

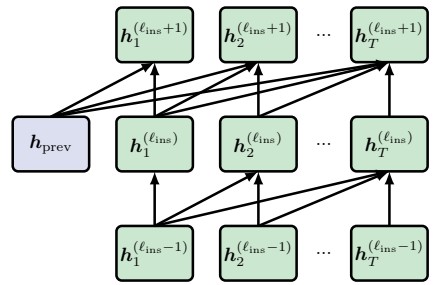

Figure 1: Augmenting a pretrained transformer with a recurrence module, allowing reduction of attention computation as well as simpler processing of longer contexts.

Figure 2: $\boldsymbol{h}_{\text{prev}}$ is added as an additional key and value to one self-attention layer. Arrows show which positions can pass information to which other positions.

the embeddings produced by the pretrained transformer are mean-pooled as follows:

$$\boldsymbol{z}_1 = \frac{1}{T} \sum_{i=1}^{T} \sum_{\ell=1}^{L} w_\ell \boldsymbol{h}_i^{(\ell)} \tag{1}$$

where $w_\ell$ are weights softmax-normalized from learned parameters $\alpha_\ell$:

$$w_\ell = \frac{\mathrm{e}^{\alpha_\ell}}{\sum\limits_{j=1}^{L} \mathrm{e}^{\alpha_j}}$$

The fixed-size representation, $\boldsymbol{z}_1$, is passed through a feedforward network to produce an embedding $\boldsymbol{h}_{\text{prev},1}$ which represents the tokens processed so far, $t_{1:T}$. Next, instead of evaluating the pretrained transformer without modification on positions $T + 1$ through $2T$, $\boldsymbol{h}_{\text{prev},1}$ is inserted at a single layer (denoted $\ell_{\text{ins}}$) of the pretrained model, as an additional embedding that may be used in the computation of attention, as shown in Figure 2. To keep the number of embeddings per layer fixed, this embedding is only used as a key and a value, but not a query, in the self-attention layer. That is, for a window size of 300 tokens, there are 301 inputs to layer $\ell_{\text{ins}}$, but still only 300 outputs. The embeddings for positions $T + 1$ to $2T$ are then pooled in the same way as Equation 1 to produce $\boldsymbol{z_2}$ and passed through the feedforward network, outputting $\boldsymbol{h}_{\text{prev},2}$. $\boldsymbol{h}_{\text{prev},2}$ is used to modify the execution of the pretrained language model on tokens $2T + 1$ through $3T$, and so on. Because the model is now being applied recurrently, it is trained end-to-end with backpropagation through time.

One could consider more complex recurrence modules, other methods for pooling the previous window's embeddings, or for inserting $\boldsymbol{h}_{\text{prev}}$ into the computation for the next window. We experimented with modifications such as max pooling instead of mean pooling, inserting multiple embeddings into the next window, inserting an embedding at all layers of the transformer for the next window, and using fixed key attention as the pooling function. However during our preliminary experiments, we were not successful in finding a significantly higher performing architecture than the one given above, so it is the one we present results for.

### 3.2 GRADIENT CHECKPOINTING IN NETWORKS WITH BOTTLENECKS

While our method can reduce the quadratic cost of attention by splitting the input into windows, we can also easily apply it to much longer contexts by use of gradient checkpointing (Chen et al., 2016).

Gradient checkpointing is a method for lowering the peak memory requirement of training large neural networks. This is accomplished by storing only a subset of activations during the forward pass, and recomputing forward from those cached states during the backwards pass. For example, in a 100 layer feedforward network with uniformly wide layers, one could store the output of only every 10th layer. Then, during the backward pass, in order to compute the gradients for the 95th layer, one would re-compute layers 91 through 99 using the stored 90th layer activations. The overall memory cost is reduced to $\sqrt{L}$ at the cost of a single additional forward pass.

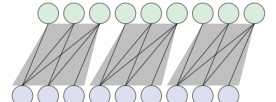 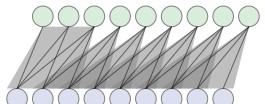 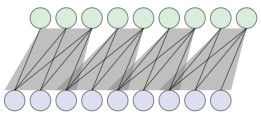

(a) Disjoint execution. Predictions have context ranging between 1 and 3 tokens.

(b) Maximum overlap. All predictions except the first two have maximal context.

(c) Intermediate degree of overlap. Except the first prediction, all predictions attend to at least 2 tokens of context.

Figure 3: Varying degree of overlap while evaluating a transformer with a window size of 3. The green (top) circles are outputs, and the blue (bottom) circles are inputs.

In a network with variable width, the memory reduction can be even larger. When gradient checkpointing is applied to transformers, the outputs of each layer are usually stored ($k \times L \times T$ values), so that at most one set of self-attention activations is in memory at once. In the case of our recurrent models, we have an even narrower bottleneck: the $z_i$'s and $h_{\mathrm{prev},i}$'s. Storing only these values means that the maximum number of activations present in memory while training on sequences $N$ tokens in length is $M + 2k\lceil\frac{N}{T}\rceil$, where $M$ is the number of activations stored when training the transformer on an individual window of length $T$. Because $k$ is extremely small compared to $M$, our model can be applied to very long contexts on any GPU on which the pretrained model can be fine-tuned.

## 4 REVISITING THE EVALUATION OF TRANSFORMER LANGUAGE MODELS

Before describing the empirical evaluation of our method, we discuss how transformer language models are evaluated in related work. The standard way of measuring perplexity uses extra computation in order to make as much context available for each token prediction. This yields low perplexities, but does not reflect how practitioners use transformer language models in applications. In this section, we describe the situation in detail and propose practical solutions that achieve relatively low perplexities while being closer to how transformers are used in practice.

### 4.1 POTENTIAL MISALIGNMENT BETWEEN LM EVALUATION AND APPLICATION

Transformers are often described as having quadratic time complexity in comparison to RNNs which have linear time complexity. However, this can be somewhat misleading when it comes to evaluation of perplexity. Given a test set of length $N$, an RNN requires $O(N)$ time to evaluate—but reaching the best perplexity for a transformer requires $O(NT^2)$, where $T$ is its maximum context size. (These time complexities exclude hidden state size, number of layers, and batch size.) This much higher time complexity is due to the fact that a transformer may be run with its full context size once for each token in the test set, so that the maximum context is available for each prediction. Re-execution of the whole model for each token is required for models with absolute position embeddings, since hidden state reuse is only possible up to the maximum context size of the network. Note that it is possible to achieve smaller wall-clock time by splitting evaluation of a test set over multiple GPUs, but this is not applicable to the generation setting where outputs depend on prior ones.

To illustrate why re-computation is necessary, consider executing GPT-2 (which has 1024 position embeddings) on a test set. Each of the first 1024 tokens of a test set will have been passed into the network using a distinct position embedding. Having exhausted the position embeddings, one option is to start again with the 1025th token being treated as position 1—we will refer to this as *disjoint execution*, illustrated in Figure 3a. The issue with disjoint execution is that it requires predicting the tokens at the beginning of a window from a very small amount of context.

The alternative, which is used for standard test set evaluation, is *overlapped execution*, as shown in Figure 3b. The position embeddings are advanced by one position for each prediction, meaning that $T-1$ tokens are repeated between consecutive evaluations of the transformer, requiring much more computation. The benefit of this method is that it allows a model with $T$ position embeddings to have $T$ tokens of context for each prediction, as opposed to a variable amount between 1 and $T$.

Stepping a transformer decoder forward one token at a time measures the best that such a model could perform, but it reflects a generative story that does not align with how the models may be used

in practice. A perplexity that only measures the ability of GPT-2 to generate the 1024th token given a context of 1023 tokens is not necessarily indicative of the model's performance when generating from a smaller context. For example, the popular website Talk To Transformer[1] generates samples from GPT-2, but only provides 150 tokens of output. The evaluation of GPT-2 by stepping forward one token at a time provides little information about the quality of such generations.

An example where the discrepancy is length instead of brevity is the GPT backed text adventure game AI Dungeon.[2] In this setting, the number of tokens can easily reach and exceed the full context size GPT-2 was trained on. Using overlapped execution as described above, generating each token would be 1024 times slower than with disjoint execution, so perplexity calculated by overlapped execution does not match this use case either.

While lower perplexity seems to correspond to better generation with shorter contexts in practice (perhaps due to parameter sharing between all sequence positions), there is no reason that this need be the case in principle. To demonstrate an extreme case of the concern being discussed, let $F$ be a transformer model with vocabulary $V$, which uses the previous 1023 tokens as context, and consider the following generative story for generating token $t_i$:

$$t_i \sim \begin{cases} \text{Uniform}(V) & \text{if } i \leq 1023 \\ F(t_{(i-1023):(i-1)}) & \text{otherwise} \end{cases}$$

Clearly the above generative model would not be of any practical use for generation or otherwise. However, because perplexity is calculated per token, increasing the size of the test set will lead to a measured perplexity that approaches that of a standard evaluation of the model $F$. This example is not representative of the models that are trained in practice, as even generations much shorter than the maximum context size from a GPT-2 model are quite impressive. However, it does demonstrate that the criteria that we use to compare models, or to select the best model during early stopping, place very high weight on the ability of the model to produce text given a full context, and a potentially vanishingly small amount on its ability to generate text using shorter contexts.

### 4.2 VARYING OVERLAP FOR EVALUATION

As we are interested in increasing computational efficiency at evaluation time for pretrained models, we investigate their performance using overlapped execution, but with a reduced degree of overlap between windows. Varying the overlap lets us investigate the connection between degree of overlap and perplexity. The overlap used in evaluation will be defined to be the number of tokens from each input window that are repeated in the next window (see Figure 3). For example, consider a window size $T = 10$ and an overlap of 3. The windows that the transformer will be executed are then $t_{1:10}$, $t_{8:17}, t_{15:24}, \ldots, t_{1+7n:10+7n}$ where $n$ indexes the window. These input windows are used to predict the spans of tokens $t_{2:11}, t_{12:18}, t_{19:25}, \ldots, t_{5+7n:11+7n}$. Figure 3c illustrates an intermediate overlap setting with $T = 3$ and an overlap of 1. The perplexity-minimizing evaluation setting is then the extreme with an overlap $T - 1$, and an overlap of 0 corresponds to disjoint execution.

While a transformer can be evaluated with any degree of overlap, our augmentation method produces the embedding $h_{\text{prev}}$, which is used during training to help predict the first token of a window. If we change the overlap at test time, the alignment of the text represented by $h_{\text{prev}}$ and the current window will be different than the model was trained for, and so performance will degrade. To address this, we use the same overlap that will be used at test time during training for the recurrent models.[3]

## 5 EXPERIMENTS

We now present experiments comparing our proposed technique to the default usage of transformer language models. We describe experiments on the WikiText-103 corpus and a subset of the PG-19

---

[1]https://talktotransformer.com/

[2]https://aidungeon.io/. Note that AIDungeon now uses the OpenAI GPT-3 API, but a similar project without OpenAI API access would still have to use GPT-2.

[3]Evaluating recurrent models trained with no overlap between adjacent windows on a different level of overlap is possible by changing which positions are pooled. We found that it led to a slight increase in perplexity, so we report results with training and evaluation matching.

corpus, using the GPT-2-small language model as the pretrained transformer in our models. We also provide proof-of-concept experiments using RoBERTa (Liu et al., 2019) on the HotpotQA (Yang et al., 2018) question answering dataset, indicating that our method can improve encoder performance for tasks other than language modeling. All of our experiments are based on the Hugging Face Transformers library (Wolf et al., 2019).

WikiText-103 is a standard language modeling corpus composed of approximately 29,000 documents from English Wikipedia, containing 103 million words. We use the WikiText-103 "raw" corpus, which does not have rare words replaced by "UNK". While GPT-2 uses BPE tokenization, we compute perplexity using the number of words rather than the number of BPE tokens for clarity.

Although WikiText-103 does test long term dependencies, many of the documents are still shorter than the context size of the models we test. Therefore, we also use PG-19, which consists of books from the Project Gutenberg corpus. The average length of a WikiText-103 document is 3.6K words, while PG-19 documents (i.e. books) average 69K words, which far exceeds the context size of the models we test. However, the full PG-19 dataset is over 20 times larger than WikiText-103, so we use only a subset of it for training due to computational constraints. Specifically, we use only the first (alphabetically by filename) 1250 books of the PG-19 corpus, and use only the first 15000 tokens of each of the books in the validation set for early stopping. We make no modifications to the test set.

In all our experiments we use the HuggingFace implementation of the pretrained GPT-2 small model (12 layers, 768-dimensional hidden state). For both the recurrent and baseline models, the GPT-2 model was fine-tuned, not left frozen. We selected learning rates for both our models and the baseline separately, by evaluating on WikiText-103 for the same set of candidate learning rates. We used the same learning rates for the PG-19 experiments without further hyperparameter search. We fine-tune all models for 2 epochs, measuring the validation loss every 2 million tokens. All models were trained with Adam (Kingma & Ba, 2014), warming the learning rate up linearly from 0 to its final value over 100 steps. The feedforward network used to produce $\boldsymbol{h}_{\text{prev},i}$ from window $i-1$ consisted of 3 hidden layers with dimension 200. We fixed $\ell_{\text{ins}}$ to be 2.[4]

Recall from Section 4 that we are interested in evaluating the models in a setting similar to how they would be used in practice. To that end, we report separate perplexities for different degrees of overlap between adjacent windows of text, as described in Section 4.2. For our models, we train with the same overlap that we test with, as unlike the baseline models, they cannot be trained with no overlap between adjacent windows and then tested with an overlap. This is because the embedding of the previous window of text is expected to represent all tokens up until the first token of the current window, but with an overlap of 30 for example, that embedding would be representing all tokens up until the 30th token of the current window.

## 5.1 RESULTS

We first show that with the same amount of fine-tuning, our method achieves lower perplexity than a baseline GPT-2 model when evaluated using the same window size and degree of overlap between adjacent windows of text.

It is important to emphasize that the perplexities we report are based on pretrained models, and so should not be compared to models trained from scratch on these datasets. The GPT-2 models were trained on text from a web crawl from which all Wikipedia documents are removed, but this still leaves open the possibility of quotes from Wikipedia having been encountered, or text from PG-19.

Table 1 shows the perplexity of our models and the non-recurrent GPT-2 models on the WikiText-103 dataset. The models compared here all use windows of 300 tokens, with varying degrees of overlap. The baseline models can only access information from the previous window of text through the overlapping tokens, while the recurrent models have a fixed size representation of the longer context. Our addition of recurrence increases the performance of the GPT-2 models in this setting, but by a relatively small amount. Increasing the overlap between each window of text decreases the perplexities of the baseline model as expected, but also decreases the perplexity of the recurrent models.[5]

---

[4]During our preliminary experiments, we found that setting $\ell_{ins}$ to be one of the final layer in the network gave slightly worse results, but we did not re-tune this hyperparameter for PG-19 or our final architecture.

[5]We did not attempt to train recurrent models with extremely high overlaps, as that would greatly increase the required training time.

Table 1: Results on WikiText-103

| Model | Overlap | Validation Perplexity | Test Perplexity | FLOPs/token |
|---|---|---|---|---|
| GPT-2 (small), 300 token window | 0 | 29.00 | 30.47 | $1.75 \times 10^8$ |
| | 5 | 27.99 | 29.36 | $1.78 \times 10^8$ |
| | 10 | 27.58 | 28.88 | $1.81 \times 10^8$ |
| | 30 | 26.72 | 27.96 | $1.94 \times 10^8$ |
| | 50 | 26.17 | 27.31 | $2.10 \times 10^8$ |
| Recurrent, 20 windows of 300 tokens (Ours) | 0 | 27.70 | 29.01 | $1.75 \times 10^8$ |
| | 5 | 26.88 | 28.12 | $1.78 \times 10^8$ |
| | 10 | 26.51 | 27.77 | $1.81 \times 10^8$ |
| | 30 | 25.90 | 27.12 | $1.94 \times 10^8$ |
| | 50 | 25.53 | 26.73 | $2.10 \times 10^8$ |

Table 2: Results on PG-19

| Model | Overlap | Validation Perplexity | Test Perplexity | FLOPs/token |
|---|---|---|---|---|
| GPT-2 (small), 300 token window | 0 | 172.25 | 147.71 | $1.75 \times 10^8$ |
| | 5 | 165.93 | 142.30 | $1.78 \times 10^8$ |
| | 10 | 162.66 | 139.49 | $1.81 \times 10^8$ |
| | 30 | 156.21 | 134.30 | $1.94 \times 10^8$ |
| | 50 | 152.64 | 131.25 | $2.10 \times 10^8$ |
| | 75 | 149.54 | 128.46 | $2.33 \times 10^8$ |
| | 100 | 147.05 | 126.51 | $2.62 \times 10^8$ |
| | 150 | 143.62 | 123.53 | $3.50 \times 10^8$ |
| | 200 | 141.14 | 121.40 | $5.25 \times 10^8$ |
| Recurrent, 20 windows of 300 tokens (Ours) | 0 | 155.27 | 133.02 | $1.75 \times 10^8$ |
| | 5 | 150.00 | 128.78 | $1.78 \times 10^8$ |
| | 10 | 147.53 | 127.05 | $1.81 \times 10^8$ |
| | 30 | 142.35 | 122.22 | $1.94 \times 10^8$ |
| | 50 | 140.10 | 119.93 | $2.10 \times 10^8$ |

This indicates that there is room to increase the capacity of the recurrence mechanism (potentially requiring more training data), as an ideal recurrence mechanism would render these overlapping tokens redundant. On the other hand, some useful information beyond what is contained in the local context is being propagated, as otherwise the baseline model should catch up in perplexity at higher overlaps. To investigate this further, we also experiment with the PG-19 dataset.

The results for the PG-19 experiments are shown in Table 2. While we find only small increases in performances on the WikiText-103 dataset, we see larger improvements on PG-19, confirming our prediction that the gains would be larger on a dataset that has a larger context available for each prediction on average. We find that adding our recurrence module leads to a model that gives as good a perplexity with no overlap between adjacent windows as an unmodified model does when evaluated with an overlap of 30 out of 300 tokens in each window. Training the recurrent model with a 5 token overlap gives perplexity lower than the baseline perplexity with an overlap of 50 or even 75. In terms of FLOPs, adding our recurrence module and overlapping adjacent windows of tokens by 50 is less than half as costly as using a non-recurrent model with an overlap of 200.

## 5.2 EFFECT OF WINDOW SIZE

As one of our motivations is to retain performance while decreasing compute requirements, we experiment with varying the window size used by our model and an unmodified GPT-2 model. At smaller window sizes the recurrent model has access to much more information than GPT-2, which can only attend to the current window. Because of this, we expect our augmentation to cause the

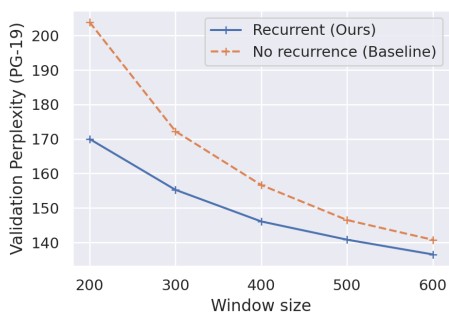

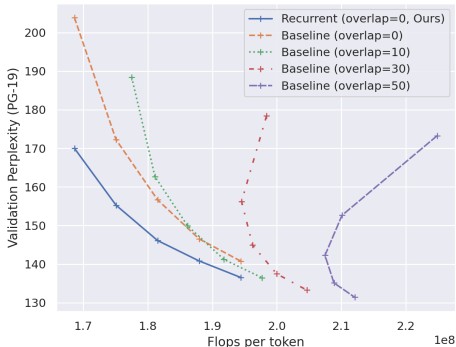

Figure 4: Effect of window size on performance on PG-19 validation set.

Figure 5: Relationship between FLOPs and perplexity. Curves range over window sizes from 200 to 600.

performance to fall off less rapidly with decreasing window size. The results, shown in Figure 4, confirm this prediction, as the performance gap widens with smaller windows. Figure 5 contains the same points (and additional baseline curves for various overlaps), but in terms of FLOPs rather than window size. All of the results of the recurrent models lie on the Pareto frontier, meaning that to improve perplexity or computational cost, one must worsen the other. The non-monotonicity of the overlap 30 and 50 curves is due to the fact that at smaller window sizes, an overlap represents a higher fraction of the computation being used for positions that predictions were already produced for. Also note that while the baseline with overlap 50 curve has the lowest absolute perplexity in Figure 5, the recurrent models trained with overlaps shown in Table 2 still perform better.

### 5.3 WHAT INFORMATION IS BEING PROPAGATED BETWEEN WINDOWS?

We now discuss some features that our models display in greedily decoded continuations from contexts in the PG-19 validation set, which illustrate types of information that the recurrent module passes (or fails to pass) forward. Samples are included in Tables 4 and 5 in the appendix.

The most common phenomenon we identify in these samples is successful propagation of topical information between adjacent windows. For instance, we see in Table 4 a context discussing geography and rivers, followed by a continuation maintaining the same topic, and we see a context discussing the topic of payment, leading to a mention of money in the continuation. We give more rigorous quantitative support of this claim in Section 5.3.1. Beyond passing of topical information, another success case in the generations is passing of certain information about characters between windows—in Table 5 we see that pronouns in the continuations often reflect characters mentioned in the context, and we see an example in which the continuation includes "the two women", after a context mentioning "the aunts". This behavior was likely learned due to the fact that PG-19 consists of narratives, so correctly passing character information between windows is quite beneficial.

However, these examples also contain discontinuities between the context and the continuation, in terms of local syntax or facts of the narrative. We see that some sentences are not completed in the expected form (for instance, "There are lots of ways of being" is continued with a new quote rather than completion of the thought), and new characters are sometimes invented rather than continuing to reference those described in the context. One sample has a closing quotation mark, predicted from the previous window, being interpreted as an opening quotation mark. These are the types of issues that an overlap between adjacent windows easily addresses—a fact that likely accounts in part for the gap between the recurrent model with disjoint and overlapped execution in Table 2. A higher capacity recurrent module might fix these issues in exchange for additional computation.

### 5.3.1 QUANTITATIVE EVALUATION OF TOPIC PROPAGATION

To verify the trend we identified of topic propagation in continuations generated by our recurrent models, we fit an LDA topic model (Blei et al., 2003) with 20 topics to 5000 books from the PG-19

Table 3: Results on HotpotQA distractor setting development set, using 30,000 randomly selected training examples. Scores are answer only (no supporting fact prediction).

| Method | F1 | Exact Match |
|---|---|---|
| Disjoint RoBERTa | 48.17 | 43.55 |
| Recurrent RoBERTa | 49.12 | 44.55 |

training set. Given a bag of words, this topic model will assign a distribution over topics, so we can use a statistical distance as a metric for the similarity between the topics of two segments of text.

We sampled 8000 contexts of 300 tokens from the PG-19 validation set, and computed argmax decoded continuations of 30 tokens from the same models used to generate Table 4[6]. We then computed the Jensen-Shannon divergence (JSD) between the topic distribution of each context and the corresponding continuations. This procedure finds that continuations from the recurrent model have an average topic JSD of 0.5331, while those from the baseline model have an average topic JSD of 0.5951. For a given context, the continuation given by the recurrent model is likely to have a lower JSD at least 60% of the time ($p < 0.00001$).

### 5.4 QUESTION ANSWERING EXPERIMENTS

To investigate whether our recurrence method would be helpful in tasks other than language modeling, we ran a small experiment on the HotpotQA extractive question answering task, in the distractor setting. In this setting, 10 paragraphs of context are given which must be used to answer the given question. HotpotQA's inputs can greatly exceed one 512 token window in length, making it an ideal test of our method. The questions are a mix of span-based and yes/no questions. In order to be able to reduce training time, we use a subset of 30000 randomly sampled questions from the training set.

We use the RoBERTa-base model for both the baseline and the recurrently augmented model. To evaluate whether recurrence improves encoder performance on this task, we directly finetune the models to predict answer span start and end tokens, as done for question answering by Devlin et al. (2019), and max pool the embedding of the [CLS] token across windows and use the result for three way classification between "span", "yes", and "no". Because the non-recurrent baseline cannot process an entire example at once, we begin each input window with the question, separated from the text by a [SEP] token. We use this input format for both the baseline and recurrent models.

For both models, we use a learning rate of 2e-5 and train for 4 epochs. For the recurrent model, we mean-pool the final RoBERTa layer for the previous window, and use a 2 layer, 768-dimensional hidden layer MLP to produce an embedding which is inserted at the second layer of the next window (i.e., $\ell_{\text{ins}} = 2$).

Table 3 shows F1 and exact match scores for both models on the HotpotQA dev set. Adding the recurrence module improves both scores by about 1 point, indicating that our method of propagating information between windows can be beneficial for question answering in addition to language modeling. It should be noted that these values are not directly comparable to scores on the HotpotQA leaderboard, as we only used a subset of the training set, in addition to evaluating on the dev set rather than the private test set.[7] Nonetheless, we find these initial experiments to be highly promising, especially given the lack of hyperparameter tuning.

## 6 CONCLUSION AND FUTURE WORK

We showed that augmenting a pretrained language model with a recurrence module during fine-tuning can allow increased performance given a fixed computational budget. Our method can be similarly applied to improve the computational efficiency of pretrained models that already exist for many languages and domains, as well as for future models that will be developed. It can also allow their application to longer contexts than they were trained for, increasing their flexibility.

---

[6]The baseline receives one token of context to begin generating from

[7]These initial results thus represent preliminary experiments that completed prior to the revision deadline; the next version of the paper will have more thorough results including when training on the entire training set.

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

## A  APPENDIX

Here we provide some example continuations for contexts from the PG-19 validation set. The samples were generated with greedy argmax decoding, which leads to a large amount of repetition, however we were more concerned with reducing variance and identifying the most likely continuation than optimizing for sample quality.

Table 4: Continuations of PG-19 validation set passages using recurrent model ($T = 300$, disjoint) which display approximate continuation of topic. Contexts are shortened for space, 300 tokens of context were used for each.

| Context | Continuation | Topic |
|---|---|---|
| ... It drains a territory of 300,000 square miles, and, traced back to the rise of its principal source, is 2,000 miles long. At two points, Needles and Yuma on the California boundary, it is crossed by a railroad. Elsewhere its course lies far from Caucasian settlements and far from the routes of common travel, in the heart of a vast region fenced on the one hand by arid plains or deep forests and on the other by formidable mountains | . The first of the two great rivers, the Mississippi, was the first to be navigated by the Americans, and the first to be navigated by the Indians. The other was the first to be navigated by the Americans, and the first to be navigated by the Indians. The first of the two great streams, the Mississippi, was navigable by the Indians, and the first by the English... | Rivers |
| ... "Mr. Hammerby will have to wait for his money." "Will he?" "I don't see what else he can do. But, aside from that, three dollars and a half won't keep us a month. I'll have to | get the money, and he'll be glad to see you again." "I'm glad to see you again, Tom. I'm glad to see you again. I'm glad to see you again. I'm glad to see you again. I'm glad to see you again | Money |
| ... Long's his sister, Thankful, lived, we couldn't say nothing, of course. If she wanted to slave and work so's her brother could live in idleness and sloth, why, that was her business. There ain't any law against a body's making a fool of herself, more's the pity. But she's been dead a year, and he's done nothing since but live on those that'll trust him, and ask help from the town. He ain't sick–except sick of work. [...] to-mor | rer and his wife, and the whole family. He was a good man, and he was a good man. He was a good man, and he was a good man. He was a good man, and he was a good man. He was a good man, and he was a good man. He | Family, Description of a man |
| ... and any form of thought or any institution tending to suppress education or destroy intelligence strikes at the very essence of the government, and constitutes a treason which no law can meet, and for which no punishment is adequate. Education, then, as universally diffused as the elements of God, is the life-blood of our body politic. The intelligence of the people is the one great fact of our civilization and our prosperity,–it | is the only way to preserve the peace of the world. The only way to preserve the peace of the world is to preserve the liberty of the people. The first step to be taken is to abolish the power of the government. The first step to be taken is to abolish the power of the people. The first step to ... | Government |

Table 5: Continuations of PG-19 validation set passages using recurrent model ($T = 300$, disjoint) which display continuation of correct pronouns or references to characters from context. Contexts are left-truncated, 300 tokens of context were given for each continuation.

| Context | Continuation |
|---|---|
| . . . Elaine, however, did not gratify their hopes; she referred to the ball with the detachment she would have shown in describing a drawing-room show of cottage industries. It was not difficult to discern in her description of the affair the confession that she had been slightly bored. From Courtenay, later in the day, the aunts received a much livelier impression of | the house, and the two women were standing in the doorway. "I am afraid I have not been very well," she said, as she looked at the two men. "I have been very much worried about you, and I have been very much worried about you. I have been very much worried about you, and I have been very much worried about you. I have been very much pleased with you, and I have been very much |
| . . . "Oh, he's not at all a bookworm," said Suzette quickly, "though he's tremendously well-read. He's quite the man of action." "Does he hunt?" asked Elaine. "No, he doesn't get much time or opportunity for riding." "What a pity," commented Elaine; "I don't think I could marry a man who wasn | " "I don't know," said the Sergeant, "but I have heard that you are a very good friend." "I am not," said the girl, "I am not." "I am not," said the Sergeant, "I am not." "I am not," said the girl, "I am not. |
| . . . Gregg was still the soft-stepping scoundrel who invited the toe of honesty, and Mornway, as he entered, was conscious of a sharp revulsion of feeling. But it was impossible to evade the interview, and he sat silent while the man stated his case. Mrs. Mornway had represented the stenographer as being in desperate straits, and ready to accept any job that could be found, but though his appearance might have seemed to corroborate her account, he evidently took a less hopeless view of his case, and the Governor found with surprise that he had fixed his eye on a clerkship | that he had been in the habit of doing, and that he had been in the habit of doing. "I'm not going to be a fool," he said, "and I'm not going to be a fool,. . . |
| . . . Many transactions too, from a desire to avoid worrying me, were carried through without my knowledge, although formerly, as a matter of course, they would have been submitted to me. Strangers, when they called, asked to see Johnson or Marsh. I directed the messenger that they were to be shown into my room if I was disengaged. This was a failure, for, when they came, I was obliged to ask for help, which was not given very generously. Sometimes I sent for the papers, but it took a long time to read them, and my visitors became impatient. During one of these interviews, I remember that I was sorely perplexed, but I had managed to | get the money, and I was glad to get it. The next day I went to the Court of the Emperor, and found him standing in the courtyard of the Palace, with a large number of soldiers. He was dressed in a very dirty uniform, and wore a very dirty hat. He was dressed in a very dirty uniform, and wore a very dirty hat. He was dressed in a very dirty hat, and wore a very dirty hat |
| . . . I know at which crossings to look out–I know what I'm going to see in the shop-windows. It saves a lot of wear and tear to know what's coming. For a good many years I never did know, from one minute to another, and now I like to think that everything's cut-and-dried, and nothing unexpected can jump out at me like a tramp from a ditch." He paused calmly to knock the ashes from his cigar, and Garnett said with a smile: "Doesn't such a plan of life cut off nearly all the possibilities?" The old gentleman made a contemptuous motion. "Possibilities of what? Of being multifariously miserable? There are lots of ways of being | "I'm afraid I'm not going to be able to do that," he said. "I'm going to have to go to the station. I'm going to have to go to the station. I want to see the station. I want to see the station. I want to see the station. I want to see the station. I want to see the station. I want to see the station. I want to see the station. I want to see |

