# OpenReview forum: " Adding Recurrence to Pretrained Transformers"
_ICLR.cc/2021/Conference — Reject_

### Official Review · AnonReviewer1 · 2020-10-26
**Interesting idea but evaluation is inadequate.**

**Rating:** 4
**Confidence:** 4

**Review:**

The goal of this work is to enable existing pre-trained transformers (e.g. GPT-2) to operate over long input contexts. This is achieved by breaking the input sequence into segments and processing each segment through the transformers while allowing tokens in the current segment to attend over a summary vector of the tokens in the previous segment. The summary vector is created as a weighted combination of the tokens in the summarized segment. Thus the summary vector introduces recurrence where each segment can use information from the previous segment. These modifications yield a better language model for long input texts.

The main benefits of this approach are as follows: (1) The modifications yield a better language model for long input texts, especially when compared to a tiling based approach (2) Potential for reducing memory footprint in these models by shrinking the amount of text that is to be processed in one-go.

My main concern with the paper is that unfortunately the evaluation is only limited to perplexity numbers on a couple of datasets. While this is a useful metric for evaluation, this alone is inadequate to demonstrate the quality of the model as a text generating system or as a language model that will be fine-tuned for target tasks or to understand how much impact the model will have in these applications.

-- For the model to be considered as a text generation system, there needs to be some human evaluation of the generated outputs. There are a small number of examples in the paper but that is not enough for a quantitative assessment. To clearly establish the benefits of the proposed modification it would be even better to consider generation tasks where conditioning on long inputs is essential.

-- How will model fare when used in a target task defined over long input contexts? The related work section includes some papers that evaluate on such tasks. For example on target task could be HotpotQA, which requires QA over ten paragraphs which easily exceed the 512 token limits. It is important to know how the proposed recurrence model compares to tiling GPT (disjoint version) or other simpler approaches on these long input tasks.

-- Another key strength of the model is that it potentially allows for processing the input in smaller segments. While perplexity gains are helpful, here again there is a missed opportunity in terms of human evaluation of the generated outputs over shorter segments, and the impact of these choices in different applications.


To reiterate, this paper presents a very nice idea to a well-motivated problem. The executed experiments show that this idea is likely to work but the gaps in experimentation leave much room for speculation about the potential impact of this approach in end applications.

---

> ### Author Response · Authors · 2020-11-25
> **Response to Reviewer 1**
>
> > My main concern with the paper is that unfortunately the evaluation is only limited to perplexity numbers on a couple of datasets. While this is a useful metric for evaluation, this alone is inadequate to demonstrate the quality of the model as a text generating system or as a language model that will be fine-tuned for target tasks or to understand how much impact the model will have in these applications.
>
> We agree that this is an important point, and have added preliminary experiments on HotpotQA as suggested. Due to the short time frame, we were only able to run a limited evaluation, using approximately one third of the training set for 4 epochs, and doing no hyperparameter search for either the baseline or our model. However, we find that adding a recurrence module to a disjoint RoBERTa baseline adds about 1 point to both the F1 and exact match scores. We find the fact that this result occurred “out of the box” without any tuning to be very promising, and will scale up the experiments to the full training set, longer training, and multiple architecture evaluations for different recurrent modules for the final version of the paper.

---

### Official Review · AnonReviewer4 · 2020-10-27
**A nice improvement for pretrained Transformers**

**Rating:** 7
**Confidence:** 5

**Review:**

Summary:

The paper proposed to add a recurrent component to pretrained transformers. The component pools the hidden states of a context window and passes it to the next context window as an additional input to the self-attention layer. The component reduces the memory usage at both training and inference time, and enables the Transformer model to work on a longer sequence. The component is evaluated on two language modeling datasets and outperforms baseline models.

Reasons for score:

I vote for accepting the paper. The paper proposed a nice and simple way to make use of the existing pretrained Transformers with reduced memory usage and extended sequence length. This should benefit practitioners who want to apply these language models on a more diverse set of downstream tasks where the sequence length doesn’t fit the one from the original pretrained model. The results presented in the paper are significant. The paper is well-written and easy to follow.

Comments:

1. It would be better to include the failure results stated in the end of section 3.1. I’m surprised that a key-value pair can boost the performance that much.
2. The paper should add more content on differentiating with Transformer-XL. I believe the difference is more than relative embeddings. For example, each Transformer-XL layer attends to an earlier layer of previous timestep, this convolutional operation making the structure no longer “recurrent”.

Typos:
- Third line in section 3.1: “at position t” -> “at position i”

---

> ### Author Response · Authors · 2020-11-25
> **Response to Reviewer 4**
>
> > It would be better to include the failure results stated in the end of section 3.1.
>
> As the failures were in our preliminary experiments, we do not have rigorous experiments for them. We can run experiments and add results on these variations in the final version of the paper.
>
> > I’m surprised that a key-value pair can boost the performance that much.
>
> If one views the output of the recurrent module as an embedding of the previous windows, it may be less surprising.
>
> > The paper should add more content on differentiating with Transformer-XL. I believe the difference is more than relative embeddings. For example, each Transformer-XL layer attends to an earlier layer of previous timestep, this convolutional operation making the structure no longer “recurrent”.
>
> We have added this to the discussion in Section 2.

---

### Official Review · AnonReviewer3 · 2020-10-28
**practically useful method for a long context size**

**Rating:** 7
**Confidence:** 2

**Review:**

The paper proposes recurrent connections between two adjacent Transformers, which transfers the previous context to the next step. This is a practically useful technique, improving the performance (perplexity in the experiments), and worth publishing. However, I have some comments and questions about the article.

Section 5.3 is an interesting question. The authors argue that topical information or so between adjacent windows is propagated. Although it is a plausible argument, it seems like it is hardly supported by table 3.

The authors said more complex recurrence modules do not make any significant difference. Then, the authors need to explain why the variations do not matter. For example, the authors fixed l_ins to be 2, without an explanation.

It is interesting to see the relationship between the overlap length and improvement using the recurrent connection. It would be better to have further discussion about the relation and different roles.

The Transformer model is also fine-tuned with the recurrent connection. So, I was wondering if the fine-tuning improves the Transformer model too. It would be interesting to compare the updated Transformer to the previous one.

In Eq. (1), is there 1/T?

---

> ### Author Response · Authors · 2020-11-25
> **Response to Reviewer 3**
>
> >  The authors argue that topical information or so between adjacent windows is propagated. Although it is a plausible argument, it seems like it is hardly supported by table 3.
>
> Based on this feedback, we have included an additional experiment (see section 5.3.1), in which we compute an LDA topic model using a portion of the PG19 training set, then compute the JS divergence between the topic distributions for a context and an argmax decoded continuation using both the baseline and recurrent models. We find that the topics for the recurrent continuations have a lower JS divergence from the context than the baseline continuations. Hopefully this additional quantitative evaluation addresses this concern.
>
> > The authors said more complex recurrence modules do not make any significant difference. Then, the authors need to explain why the variations do not matter.
>
> To be clear, we were not saying that there do not exist better module architectures, but that we were unable to find any, and so went with the simplest architecture that gave good results. We have rephrased the wording at the end of Section 3.1 to better reflect this.
>
> > For example, the authors fixed l_ins to be 2, without an explanation.
>
> We did not perform hyperparameter search for this value for our final experiments, and had just left it fixed since our preliminary experiments. We have added a footnote to Section 5 clarifying this. Because we did not vary this hyperparameter in our final experiments, it could have some impact, but unfortunately we lack the computational resources to perform thorough hyperparameter tuning. Nonetheless, we will include more experimental comparisons of hyperparameters and architectures in the next version.
>
> > It is interesting to see the relationship between the overlap length and improvement using the recurrent connection. It would be better to have further discussion about the relation and different roles.
>
> Could you provide more detail on what you mean by different roles? Perhaps you mean determining what kinds of improvements are achieved by overlap vs by the recurrent connection? We suspect using overlap with the recurrent module would cause the latter to focus more on longer-distance information, but we would have to verify that with further analysis.
>
>
> > The Transformer model is also fine-tuned with the recurrent connection. So, I was wondering if the fine-tuning improves the Transformer model too. It would be interesting to compare the updated Transformer to the previous one.
>
> Interesting point. If we understand correctly, you are interested in knowing whether the transformer is getting better due to the presence of the recurrent component during fine-tuning even if we don't use the recurrent module at test time. Even when using the recurrent module, the first window in each document is processed without any information from prior windows, and we could indeed evaluate our model in a way in which we treat every window as the initial window in the document. This would hopefully address your question. We will work on including that evaluation in the next version.

---

### Author Response · Authors · 2020-11-25
**Overall response**

Thank you to each of the reviewers for the helpful comments. We have revised the paper, making some points more clear, and also added two new experiments to address specific concerns which were raised.

The more substantial experiment is on the HotpotQA task, although due to time constraints we give preliminary results. We are encouraged by the fact that our method improves over a non-recurrent baseline out of the box, with no hyperparameter tuning or architecture search. For the final version of the paper, we will scale up these experiments to the entire dataset, as well as optimizing hyperparameters for both the baseline and recurrent versions, and doing some recurrent architecture experimentation.

In order to address Reviewer 3’s concern about how well our claims about topical propagation between windows were supported, we have added quantitative evidence using an LDA topic model fit to the PG-19 training set. We showed that the JS-divergence between distributions of topics computed from a context and continuations of that context are lower for our recurrent models than from a baseline continuation.

---

### Decision · Program_Chairs · 2021-01-07
**Final Decision**

**Decision:**

Reject

**Comment:**

In this paper, the authors propose to add recurrence to pre-trained language models such as GPT-2 or BERT. The idea is similar to the compressive transformer paper: a small module is added to the network, and used to compress the representations from the previous chunk of data from the sequence to a single vector. Then, this vector is added to the keys and values of the self-attention module when processing the next chunk. The main contribution of the paper is to show that this technique can be added to pre-trained models at fine-tuning time.  The main concerns regarding the paper are technical novelty and limited empirical results. The idea of adding recurrence to transformers was previously explored in compressive transformer, and many previous work have considered adding modules with small number of parameters at fine-tuning time. Moreover, I do not believe that the empirical section is strong enough to justify the acceptance of the paper, as the method is only evaluated on two language modeling tasks (and one early experiment on HotpotQA). The baselines are weak, and thus, the results are not convincing. For these reasons, I weakly recommend to reject the paper, and encourage the authors to make the empirical section stronger.